# CONTEXTKEEPER: HEAD-SPECIFIC KV CACHE RETENTION FOR LONG-CONTEXT LLM INFERENCE

## ABSTRACT

Large Language Model (LLM) inference commonly requires caching all Key-Value (KV) states. This KV cache leads to substantial memory usage and increasing latency in long-context settings. Existing KV cache compression methods reduce cache size by keeping only tokens relevant to the current query, but discarding middle context tokens needed by queries in later turns - harming multi-turn fidelity. We observe head specialization: a minority of attention heads are Context-Anchored (CA), preferring middle context tokens, while most are locality heads, focusing on sink tokens and recent tokens. This motivates ContextKeeper, a training-free, head-specific KV retention policy that preserves all middle context tokens for CA heads and drops them for locality heads. Unlike prior head-splitting methods that require complex training procedures or deliver limited gains, our policy is derived by running inference on a small set of task samples and integrates as a plug-and-play inference strategy. ContextKeeper reduces KV cache size by up to 3.86× and lowers decoding latency by up to 1.25×, while introducing negligible accuracy loss compared to full attention across different models and 5-turn queries with up to 128K tokens. These results demonstrate a practical and scalable query-agnostic KV compression method that preserves multi-turn fidelity under tight memory budgets for long-context deployment.

## 1 INTRODUCTION

The maximum context length of LLMs has been rising rapidly: models like Llama-3 (Dubey et al., 2024) and Qwen-3 (Bai et al., 2023a) support 128K tokens, and Gemini-2.5 (Comanici et al., 2025) reaches up to 1M. This unlocks applications such as multi-turn dialogue (Li et al., 2025), long-document summarization (Zhang et al., 2024) and repository-level code reasoning (Ma et al., 2024). However, autoregressive decoding requires each new token to attend to all previous ones, so all Key–Value (KV) states must be cached. As input length grows, the KV cache grows linearly, pushing memory limits and slowing inference.

A common strategy is KV cache compression: keeping only the tokens considered important for the current query and discarding others. Token-level methods such as StreamingLLM (Xiao et al., 2023) and SnapKV (Li et al., 2024b) follow this pattern. While effective in the short term, they drop middle context tokens that later queries may need, weakening multi-turn fidelity. Head-level methods build on the idea of "retrieval heads" (Wu et al., 2024) —attention heads that can copy or retrieve information from the prompt. Head-KV (Fu et al., 2024) exploits this by allocating more cache budget to such heads, improving accuracy at fixed compression ratios, but still discards middle context tokens for other heads. DuoAttention (Xiao et al., 2024) and RazorAttention (Tang et al., 2024) go further, keeping full caches for these heads and shortening caches elsewhere. DuoAttention requires optimization on synthetic data, while RazorAttention achieves more modest compression even with compensation tokens. Across these works, "retrieval head" is loosely defined, and detection methods differ.

We segment tokens into **S**ink (first $s$ tokens), **C**ontext (middle tokens), and **R**ecent (last $r$ tokens), formalized in Section 3.2. And we observe a distinct specialization pattern: only a minority are Context-Anchored (CA), consistently attending to **C**, while most are locality heads, focusing on **S** and **R**. Preserving all middle context tokens is therefore only necessary for the CA heads.

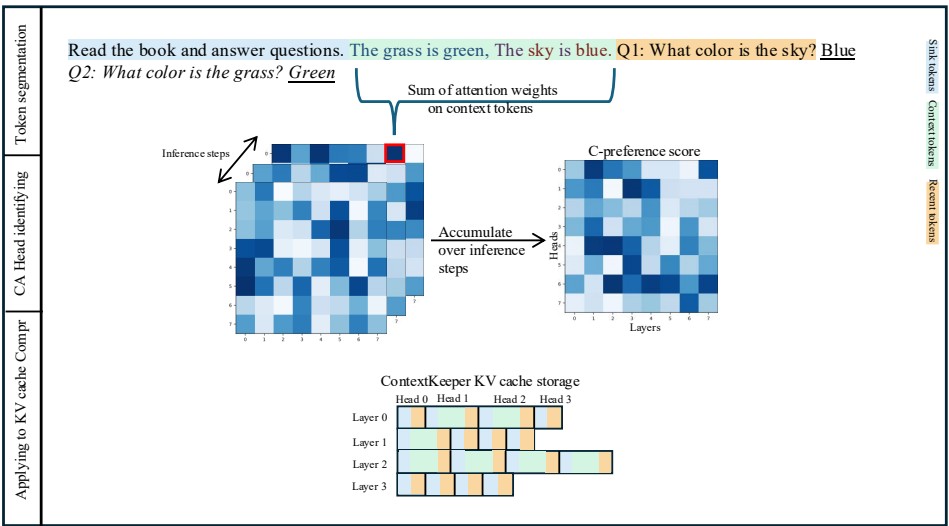

Figure 1: Overview of ContextKeeper: identifying Context-Anchored (CA) heads and applying head-specific KV cache retention. Input tokens are segmented into sink (S), middle context (C), and recent (R) tokens. Attention weights on C tokens are accumulated across inference steps to compute a C-preference score per head. And final CA heads are obtained by voting across samples from diverse tasks. The storage diagram illustrates (with 4 layers and 4 heads as an example) that CA heads retain the full KV cache, while other heads keep only S and R tokens.

Building on this observation, we propose ContextKeeper, a training-free, head-specific KV retention policy. ContextKeeper preserves all middle context tokens for CA heads while dropping them for locality heads. Unlike prior head-splitting methods that require complex training or deliver limited gains, ContextKeeper identifies CA heads by running inference on a small set of task samples, accumulating per-head attention, and selecting the heads most anchored to context. The resulting policy integrates as a plug-and-play inference strategy and is compatible with FlashAttention (Dao, 2023).

We evaluate ContextKeeper on models with Multi-Head Attention (MHA) and Grouped-Query Attention (GQA) across tasks including string and semantic retrieval, single-document and multi-document QA, repository-level code QA, summarization, and few-shot learning, with up to 128K tokens and 5-turn dialogues. ContextKeeper reduces KV cache size by up to 3.86× and lowers decoding latency by up to 1.25×, while maintaining accuracy close to full-cache baselines. Crucially, unlike query-aware methods, it preserves multi-turn fidelity, correctly answering follow-up queries that depend on earlier context.

Our contributions are as follows:

- We propose ContextKeeper, a training-free, head-specific KV retention method for long-context LLM inference.
- We introduce the notion of context-anchored (CA) heads, distinct from prior "retrieval heads," and design a lightweight, inference-only identification procedure.
- We demonstrate substantial memory and latency savings with negligible accuracy loss, and provide extensive evaluation (up to 128K tokens, 5-turn dialogues) showing preserved multi-turn fidelity.

## 2 RELATED WORK

### 2.1 ATTENTION HEADS IN MECHANISTIC INTERPRETABILITY

Mechanistic interpretability (Olsson et al., 2022) work characterizes what attention heads do, such as induction, echo (Olsson et al., 2022) and retrieval (Wu et al., 2024) patterns (see a recent survey

(Zheng et al., 2025) on attention-head functions and methodologies). Although many papers refer to "retrieval heads", the term lacks a single, agreed definition and detection criteria vary - some emphasize span copying, others induction-like and echo patterns. We adopt a complementary lens: we define Context-Anchored (CA) heads as heads that exhibit a stable preference for middle context tokens (C) in typical prompts, a property that better predicts whether information from earlier turns should be preserved. In practice, CA heads subsume strict induction and echo behaviors when they arise, and they also capture broader "Content Gatherer Head" (Lieberum et al., 2023; Merullo et al., 2023) that collects relevant information needed to the produce the final answer tokens.

## 2.2 KV CACHE COMPRESSION

A large family of query-aware methods compresses the KV cache for the current query. Token-level selection (e.g., StreamingLLM (Xiao et al., 2023), SnapKV (Li et al., 2024b)) uses attention-derived signals or the observation window to keep only the tokens considered important for the current query while dropping middle context tokens that later queries may need. Head-budgeting approaches (e.g., Head-KV (Fu et al., 2024), Ada-KV (Feng et al., 2024)) allocate larger cache budgets to heads believed to be retrieval-heavy, improving accuracy at a fixed compression ratio. These designs improve short-term efficiency, but because they are current-query centric, they tend to discard context tokens that later turns require, leaving multi-turn fidelity fragile.

Another line retains full caches only for a subset of heads. DuoAttention (Xiao et al., 2024) keeps full KV for heads identified as "retrieval" and shortens caches for the rest, using optimization on synthetic data to select those heads. RazorAttention (Tang et al., 2024) adopts a similar split, designating "retrieval heads" via induction and echo behaviors and introducing a compensation token to mitigate information loss. These methods demonstrate that head specialization can be exploited, but the head notion itself is not standardized and identification often requires auxiliary mechanisms or yields more modest compression ratio. In contrast, ContextKeeper defines CA heads by their stable context tokens preference and identifies them with a lightweight inference-only procedure. The resulting policy explicitly protects information that follow-up queries depend on.

System-level and quantization techniques are orthogonal and compatible with retention strategies. FlashAttention (Dao, 2023) reduces attention compute and memory traffic but does not shrink the KV-cache size. KV-cache quantization (Hooper et al., 2024) reduces KV memory footprint but does not reduce attention computation. ContextKeeper is compatible with both and can be used in combination to realize additive gains.

In summary, prior work either prioritizes the current query and risks erasing future-critical context tokens, or relies on head-categories and discovery procedures that are not aligned with multi-turn needs. ContextKeeper fills this gap with a training-free, head-specific retention policy grounded in CA heads, preserving context where it matters while achieving substantial KV reductions.

## 3 METHOD

### 3.1 FROM RETRIEVAL HEADS TO CONTEXT-ANCHORED HEADS

**Drawbacks of retrieval heads** Prior work (Wu et al., 2024; Tang et al., 2024; Xiao et al., 2024) often refers to retrieval heads, but the notion is not unified and is typically tied to specific surface-level behaviors such as induction or echo patterns. These heads are usually detected with synthetic string-retrieval datasets, which limits the scope of what they capture. Such detection pipelines cannot generalize to higher-level semantic behaviors, for example in summarization tasks where a model abstracts information from long passages. As shown in Figure 2, the model may allocate attention to what prior work (Zheng et al., 2025) calls [END] tokens - boundary tokens such as punctuation (".", ")") or frequent function words ("that", "the", "and") - which play an essential role in gathering information for the final summary. Existing retrieval head definitions, focused on induction or echo patterns, are unable to detect such heads even though they are critical for higher-level reasoning.

**Context-Anchored Heads** To address these limitations, we propose a new head category called Context-Anchored (CA) heads, designed with three properties. First, they are task-general, capturing not only low-level retrieval behaviors such as induction and echo but also higher-level semantic and boundary-focused behaviors. Second, they support follow-up queries by anchoring on context

```
<|begin_of_text|><|start_header_id|>user<|end_header_id|>ĊĊ
You are given a report by a government agency. Write a one-page summary of the report.
Report:
Medicare is a federal program that pays for covered health care services of qualified beneficiaries It was established in 1965 under Title XVIII of the Social
Security Act to provide health insurance to individuals 65 and older, and has been expanded over the years to include permanently disabled individuals under
65. The program is administered by the Centers for Medicare & Medicaid Services (CMS), within the U.S. Department of Health and Human Services (HHS).
```

Figure 2: Example of a context-anchored (CA) head in Llama-3.1-8B-Instruct (layer 14, head 30) on the GovReport summarization task. Attention weights on prompt tokens are visualized with a plasma colormap, where warmer colors indicate stronger attention. This head consistently attends to sentence-boundary [END] tokens, which play a crucial role in gathering and organizing information for the final summary. Such boundary-focused, higher-level behavior cannot be captured by simple low-level induction or echo patterns, underscoring the distinction between CA heads and previously defined "retrieval" heads.

tokens that later turns are likely to require, thereby preserving multi-turn fidelity. Third, they are practically identifiable, using a lightweight and training-free inference procedure rather than optimization or retraining.

Formally, we segment prompts into sink, context and recent tokens (formal definitions in Section 3.2) and define a context-anchored head as:

> *an attention head that consistently prefer context tokens over sink or recent tokens across diverse tasks, making it essential to preserve these tokens for accurate follow-up queries.*

## 3.2 IDENTIFYING CONTEXT-ANCHORED HEADS

**Token segmentation (S/C/R)** Let a prompt of length $T$ be partitioned into: Sink tokens (S) - the first $s$ tokens (including BOS and any anchor/sink tokens) that often attract attention weights (Xiao et al., 2023); Recent tokens (R) - the final $r$ tokens that contains the current user query/instruction (Li et al., 2024b); And Context tokens (C) - all remaining middle tokens, so that $T = s + c + r$ and $C = \{s + 1, ..., T - r\}$.

**Detection principle** By definition, CA heads prefer C: they allocate more attention to context tokens than S or R and do so consistently across tasks. Our identification flow therefore (i) measures per-head attention to C during both prefill and early decoding, and (ii) selects the heads that repeatedly exhibit strong C-preference.

**Profiling set** We build a small profiling set $\mathcal{D}$ spanning string retrieval, semantic retrieval, and summarization, ensuring coverage beyond induction / echo patterns. For each sample $x \in \mathcal{D}$, we run one inference pass and collect attention to C in two places: (a) prefill - restricted to an observation window of last $m$ attention rows (as in SnapKV (Li et al., 2024b)) to focus on query-to-context interactions; and (b) decoding - for the first $K$ generated steps, where follow-up behavior emerges. This keeps memory low while capturing the attention weights that matter for identification.

**Scoring and selection (per sample)** For each layer/head $(\ell, h)$ and sample $x$, we aggregate attention weights to C over the prefill observation window and the first $K$ decoding steps, yielding a C-preference score $s^{(\ell,h)}(x)$. Within each layer, we select the top-$p\%$ heads by $s^{(\ell,h)}(x)$ as that sample's CA candidates. See Figure 1 for an illustration of this process.

Mathematically, let $A_{t,i}^{(\ell,h)}$ be the attention from step $t$ to token $i$. For sample $x$ with context tokens index set $\mathbb{C}_x$, define

$$s^{(\ell,h)}(x) = \frac{1}{|U_x| + |V_x|} \left( \sum_{t \in U_x} \sum_{i \in \mathbb{C}_x} A_{t,i}^{(\ell,h)} + \sum_{t \in V_x} \sum_{i \in \mathbb{C}_x} A_{t,i}^{(\ell,h)} \right), \tag{1}$$

where $U_x$ are the last $m$ prefill rows and $V_x$ the first $K$ decoding steps.

**Consensus across samples and tasks** We then vote across the profiling set: a head $(\ell, h)$ is retained if it satisfies both a task-level consensus of at least $t\%$ (present in that fraction of tasks) and a

sample-level consensus of at least $s\%$ (present in that fraction of samples). The surviving heads are our final CA heads. This training-free procedure is lightweight, stable across diverse tasks, and directly aligned with the CA head definition.

### 3.3 APPLYING CA HEADS TO KV CACHE COMPRESSION

**Retention policy** We split all attention heads into two groups according to the identification procedure. For CA heads, we retain the full KV cache, so that all information needed for follow-up queries is preserved. For locality heads, we retain only sink tokens and recent tokens, while dropping context tokens. Unlike prior work, we allow the recent token window to grow with decoding, since generated tokens may later be referenced in follow-up queries.

**MHA and GQA models** For models with multi-head attention (MHA), the policy is applied directly at the level of KV heads. For grouped-query attention (GQA), where multiple query heads share a KV group, we apply the policy at the group level: we compute the mean C-preference score of the heads in each group and select CA groups accordingly.

**Plug-and-play deployment** Because different heads retain different token subsets, efficient KV storage is required. We adapt the flattened KV storage design from AdaKV (Feng et al., 2024), which stores each head's KV tensors contiguously in memory, and we invoke the variable-length FlashAttention API to compute attention efficiently. This makes ContextKeeper fully FlashAttention-compatible and practical for deployment.

## 4 EXPERIMENTS

### 4.1 EXPERIMENT SETTINGS

**Models and datasets** We evaluate ContextKeeper on two open-source LLMs: Llama-2-7B-32K-Instruct (Together, 2023), a multi-head attention (MHA) long-context variant of Llama-2-7B (Touvron et al., 2023) that extends the context length from 4K to 32K, and Llama-3.1-8B-Instruct (Dubey et al., 2024), a grouped-query attention (GQA) model supporting up to 128K tokens. For benchmarks, we use LongBench (Bai et al., 2023b) and SCBench (Li et al., 2024a). LongBench covers tasks such as single-document QA, multi-document QA, summarization, few-shot learning, synthetic tasks, and code completion, with input lengths ranging from 4K to 20K tokens. For SCBench, we select nine representative tasks, belonging to string retrieval, semantic retrieval, and global information processing categories, with average lengths between 22K and 198K tokens. Inputs are truncated to 128K tokens to respect model limits. To evaluate multi-turn performance, we use the multi-turn mode of SCBench, where later queries are continuously appended to the current conversation. For multi-turn, following SCBench, we append ground-truth answers from previous turns to the prompt (we do not use model self-generations).

**Baselines** We compare against three strong KV-cache compression methods: SnapKV (Li et al., 2024b), RazorAttention (Tang et al., 2024), and DuoAttention (Xiao et al., 2024).

- SnapKV selects important tokens based on attention weights from the final segment of the prompt, keeping only those for the current query. Other token-level methods such as Ada-KV (Feng et al., 2024) and Head-KV (Fu et al., 2024) may improve accuracy at a fixed compression ratio but still discard tokens needed by later queries, harming multi-turn performance.
- RazorAttention defines retrieval heads as those exhibiting induction and echo patterns, and retains the full cache for these heads, while keeping only sink tokens and rolling recent tokens for others.
- DuoAttention identifies retrieval heads via an optimization-based neural classifier trained on a passkey-retrieval dataset, then retains the full cache for selected heads and sink tokens plus rolling recent tokens for the rest.

We implement or use public source code for baselines whenever available, verifying outputs against reported results. Hyperparameter settings for each baseline are described in the following subsections.

| Models | Sink tokens | Recent tokens | Top-$p$ (fraction) | Sample consensus | Task consensus |
|--------|-------------|---------------|--------------------|------------------|----------------|
| Llama-2 | 128 | 256 | 80 - 85 | 0.8 - 0.9 | 0.8 - 0.9 |
| Llama-3.1 | 128 | 256 | 60 - 70 | 0.8 - 0.9 | 0.8 - 0.9 |

Table 1: Hyperparameters for CA heads identification.

**Implementation details** We implement ContextKeeper in PyTorch using the Transformers (Wolf et al., 2020) library. KV caches are stored with a flattened KV design adapted from Ada-KV (Feng et al., 2024), and attention is computed with the variable-length FlashAttention API (Dao, 2023). All experiments use greedy decoding, float16 precision, and a single NVIDIA H100 GPU.

**CA head identification** To build the profiling set for head detection, we select tasks from string retrieval, semantic retrieval, and summarization: Key-Value Retrieval (Liu et al., 2023), Qasper (Dasigi et al., 2021), HotpotQA (Yang et al., 2018), and GovReport (Huang et al., 2021). Note that samples overlapping with LongBench are excluded to prevent leakage. A summary of hyperparameters for CA heads identification is provided in Table 1, and we select hyperparameters on the profiling set only and use the same setting for all test tasks without further tuning.

## 4.2 LONGBENCH EVALUATION

We evaluate all LongBench tasks, which span single-document QA, multi-document QA, summarization, few-shot learning, synthetic tasks, and code completion. For SnapKV, we set the observation window size to 256 and use a KV budget of 4,096 tokens. For head-splitting methods, including RazorAttention, DuoAttention, and ContextKeeper, we retain the full cache for 25% of heads on the Llama-2-7B-32K-Instruct model and 50% of heads on the Llama-3.1-8B-Instruct model, while keeping 128 sink tokens and 256 recent tokens for the remaining heads. Because the average input length of LongBench tasks is around 8K tokens, these settings result in comparable KV footprints across all methods, ensuring a fair comparison.

**Results and analysis** The results are summarized in Table 2 and 3. On the Llama-2-7B-32K-Instruct model, ContextKeeper achieves an average score of 39.47 compared to 40.40 with the full cache, corresponding to negligible accuracy degradation while reducing the KV cache size by nearly 4×. On the Llama-3.1-8B-Instruct model, the average score with ContextKeeper is 49.45, essentially identical to the full-cache baseline of 49.43. These results confirm that ContextKeeper maintains overall accuracy on LongBench while providing substantial compression benefits.

Beyond the averages, several task-level patterns are worth highlighting. ContextKeeper performs especially well on tasks that require global information processing. For example, on the GovReport summarization task, it achieves 34.72 on Llama-3.1-8B, outperforming RazorAttention (32.14) and DuoAttention (33.66). This demonstrates that the identified CA heads not only subsume low-level retrieval behaviors such as induction and echo but also capture higher-level semantic patterns necessary for summarization. Gains are also evident on multi-hop QA tasks: on MuSiQue, ContextKeeper obtains 22.47 on Llama-2-7B and 36.54 on Llama-3.1-8B, the best results among all compared methods. And on 2wikiMQA with Llama-2-7B, it improves over both RazorAttention and DuoAttention. These outcomes indicate that ContextKeeper preserves information in contexts that span multiple reasoning steps, which is critical for complex question answering.

There are also a few tasks where alternative methods slightly outperform ContextKeeper. On SAM-Sum with Llama-2-7B, SnapKV achieves the highest score, which may reflect its strong bias toward recent tokens in short conversational summaries. On HotpotQA and MF-en with Llama-3.1-8B, RazorAttention achieves marginally higher scores than ContextKeeper, while DuoAttention slightly surpasses ContextKeeper on TriviaQA. However, these differences are small and remain within the margin of near-full performance, whereas the broader trend favors ContextKeeper across a diverse range of tasks.

Finally, we note that head-splitting policies interact differently with grouped-query attention. In Llama-3.1-8B, four query heads share one KV group, so retention must be applied at the group level. This granularity means that keeping a single CA head effectively preserves the full cache

| Tasks | Full KV | SnapKV(4K) | Razor(25%) | Duo(25%) | Context(25%) |
|---|---|---|---|---|---|
| NrtvQA | 23.95 | 21.91 | 20.12 | 20.49 | **23.84** |
| Qasper | 31.87 | 26.10 | 26.52 | 26.59 | **30.50** |
| MF-en | 41.07 | 35.30 | 36.14 | 25.49 | **42.92** |
| HotpotQA | 50.40 | 44.17 | 46.92 | **50.44** | 49.86 |
| 2wikiMQA | 34.18 | 34.23 | 34.01 | 33.37 | **35.10** |
| MuSiQue | 22.23 | 20.49 | 19.86 | 19.27 | **22.47** |
| GovReport | 33.65 | 22.40 | 25.71 | 27.98 | **30.19** |
| QMSum | 22.86 | 21.54 | 21.09 | 21.48 | **22.97** |
| MultiNews | 26.79 | 25.16 | 25.11 | 25.03 | **25.51** |
| TREC | 71.50 | 65.00 | 65.00 | 68.50 | **72.00** |
| TriviaQA | 86.09 | 83.98 | 84.01 | **86.15** | 85.89 |
| SAMSum | 42.26 | **39.84** | 32.49 | 33.10 | 37.93 |
| PCount | 0.00 | 0.00 | 0.00 | **0.33** | 0.00 |
| PRe | 53.92 | 36.25 | 46.23 | 47.25 | **49.00** |
| LCC | 52.10 | 49.21 | 48.91 | 48.34 | **51.86** |
| RB-P | 53.49 | 48.75 | 48.05 | 48.58 | **51.59** |
| **Average** | 40.40 | 35.90 | 36.26 | 36.40 | **39.47** |

Table 2: Experiment results across tasks on LongBench with Llama-2-7B-32K-Instruct model.

| Tasks | Full KV | SnapKV(4K) | Razor(50%) | Duo(50%) | Context(50%) |
|---|---|---|---|---|---|
| NrtvQA | 32.02 | 30.71 | 31.35 | 31.71 | **32.07** |
| Qasper | 48.38 | 46.02 | 46.47 | 47.29 | **47.51** |
| MF-en | 56.53 | 55.03 | **56.30** | 56.29 | 56.16 |
| HotpotQA | 57.33 | 56.99 | **57.76** | 57.05 | 57.60 |
| 2wikiMQA | 49.13 | 48.76 | 49.02 | 48.82 | **49.12** |
| MuSiQue | 36.01 | 34.90 | 36.26 | 36.16 | **36.54** |
| GovReport | 34.59 | 27.98 | 32.14 | 33.66 | **34.72** |
| QMSum | 24.91 | 24.40 | 24.33 | **25.16** | 24.99 |
| MultiNews | 27.33 | 26.26 | 27.11 | 26.88 | **27.14** |
| TREC | 69.00 | 66.00 | 69.00 | 69.00 | **69.00** |
| TriviaQA | 91.96 | 91.93 | 91.63 | **92.02** | 91.72 |
| SAMSum | 33.13 | 32.20 | 31.17 | **32.65** | 32.48 |
| PCount | 10.75 | 11.00 | 11.00 | 11.00 | **11.00** |
| PRe | 100.0 | 99.50 | 99.59 | 99.50 | **99.50** |
| LCC | 64.99 | 63.23 | 63.45 | **65.58** | 65.16 |
| RB-P | 54.90 | 51.43 | 53.36 | 55.44 | **56.50** |
| **Average** | 49.43 | 47.90 | 48.75 | 49.26 | **49.45** |

Table 3: Experiment results across tasks on LongBench with Llama-3.1-8B-Instruct model.

for the entire group, increasing the overall retention fraction to 50% for all head-splitting methods. Even under this stricter setting, ContextKeeper consistently provides the strongest or near-strongest performance across tasks, demonstrating the robustness of our identification and retention strategy.

### 4.3 SCBENCH

We evaluate multi-turn performance on SCBench, whose inputs are substantially longer than Long-Bench (often tens to hundreds of thousands of tokens, with some sequences approaching 198K). Because Llama-2-7B-32K-Instruct is limited to 32K context, we report results only for Llama-3.1-8B-Instruct. Hyperparameters follow the LongBench setup: for SnapKV we use an observation window of 256 and a fixed 4096-token KV budget. For head-splitting methods (RazorAttention, DuoAttention, ContextKeeper) we retain full KV for 50% of heads, and for the remaining heads we keep 128 sink tokens and 256 recent tokens. Because SCBench sequences are much longer, head-splitting methods naturally have larger KV footprints than SnapKV under these representative budgets.

**Results and analysis** We report multi-turn average score in the Table 4. On average, ContextKeeper achieves 46.49 versus 52.31 for full KV—about 89% of full performance and outperforms the other

| Tasks | FullKV | SnapKV(4K) | Razor(50%) | Duo(50%) | Context(50%) |
|---|---|---|---|---|---|
| Retr.KV | 83.60 | 0.00 | 9.80 | 59.00 | **68.40** |
| Retr.Pre-Suf | 58.20 | 0.40 | 2.60 | 39.65 | **43.21** |
| Retr.MultiHop | 52.40 | 22.44 | 34.98 | 37.24 | **49.91** |
| Code.RepoQA | 52.05 | 0.91 | 33.26 | 44.32 | **48.64** |
| En.QA | 29.57 | 12.17 | 26.45 | 26.67 | **29.56** |
| En.MultiChoice | 74.69 | 49.27 | 60.16 | 66.76 | **69.79** |
| Math.Find | 38.80 | 22.60 | **34.60** | 25.87 | 28.20 |
| ICL.ManyShot | 40.37 | **68.52** | 39.63 | 39.63 | 38.52 |
| En.Sum | 41.10 | 29.03 | 40.51 | 41.02 | **42.22** |
| **Average** | 52.31 | 22.82 | 31.33 | 42.24 | **46.49** |

Table 4: Experiment results across retrieval, code, QA, math, and summarization tasks on SCBench with Llama-3.1-8B-Instruct model.

compression methods by a substantial margin (DuoAttention 42.24; RazorAttention 31.33; SnapKV 22.82).

Beyond the average, the gains are broad: ContextKeeper is best among compression methods on 7 of 9 tasks. In particular, it leads across all three retrieval categories—Retr.KV (68.40), Retr.Prefix-Suffix (43.21), and Retr.MultiHop (49.91)—and on Code.RepoQA (48.64). It also matches or exceeds full-KV quality on several language tasks: En.QA is essentially identical to full (29.56 vs 29.57), En.MultiChoice narrows the gap (69.79 vs 74.69 full), and En.Sum slightly surpasses full KV (42.22 vs 41.10), consistent with the intuition that preserving context on CA heads supports global information aggregation in multi-turn dialogue.

There are two notable exceptions. First, ICL.ManyShot strongly favors SnapKV (68.52), which is unsurprising: aggressive retention of the most recent tokens is advantageous when many exemplars appear near the end of the prompt and middle-context content functions primarily as noise. Second, Math.Find is led by RazorAttention (34.60), suggesting that its induction/echo-based selection can help on pattern-localization tasks. In both cases, however, ContextKeeper remains competitive elsewhere and delivers the highest average across the benchmark.

## 4.4 EFFICIENCY RESULTS

We evaluate the efficiency of ContextKeeper on Llama-2-7B-32K-Instruct (MHA) and Llama-3.1-8B-Instruct (GQA). Following the previous sections, we retain full cache for 25% of heads on Llama-2 and 50% of KV-groups on Llama-3.1, ensuring accuracy close to the full-KV baseline. We measure three aspects: (i) the cost of identifying CA heads, (ii) peak GPU memory usage at different context lengths, and (iii) decoding latency with different generation lengths. Peak memory is reported by nvidia-smi as the maximum usage during each run. Latency is measured as end-to-end decoding time (excluding prefill) with a 32K-token prefill, generation lengths of 128–1024 tokens, batch size 1, greedy decoding, float16 precision, on a single H100 GPU.

**Results and analysis** Profiling CA heads requires only a few inferences on the small task set, taking about 10 minutes on a single GPU. By contrast, DuoAttention trains a neural network with an optimization-based procedure. Even with tuned hyperparameters, it requires about 4 hours on 8×A100 GPUs, highlighting the practical advantage of ContextKeeper's training-free identification.

Figure 3 reports peak memory across context lengths. For Llama-2-7B-32K, the base model itself occupies 13,379 MiB, and the remainder is dominated by the KV cache. At the maximum supported context length of 32K tokens, ContextKeeper reduces KV memory from 28,525 MiB (full KV) to 7,392 MiB, achieving a 3.86× reduction. This saving allows the model to process sequences it otherwise could not: the full-KV baseline runs out of memory at 128K tokens, while ContextKeeper completes inference at 45,118 MiB. For Llama-3.1-8B, the base model occupies 15,971 MiB. Because KV retention must be applied at the group level, keeping one CA head preserves the entire group, resulting in a higher effective keep-fraction (50%). Consequently, memory reductions are more modest but still consistent: up to 1.26× at 128K tokens.

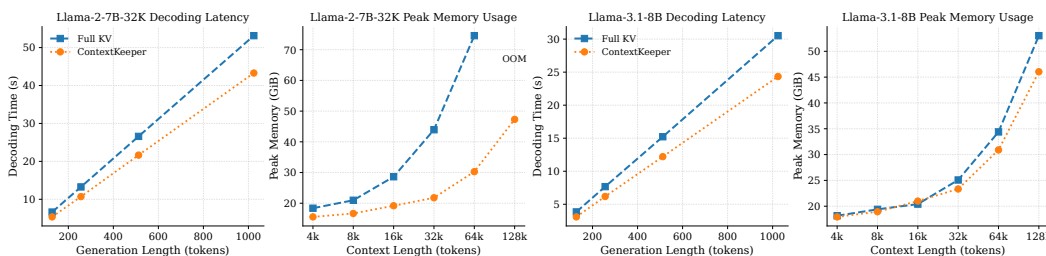

Figure 3: The decoding latency and peak memory usage results with Llama-2-7B-32K-Instruct and Llama-3.1-8B-Instruct model.

Figure 3 also presents decoding time for generation lengths of 128–1024 tokens after a 32K prefill. ContextKeeper achieves a 1.25× speedup across both models. For example, on Llama-2-7B-32K, generating 1024 tokens takes 43.3s with ContextKeeper versus 53.1s with full KV. Similar improvements are observed on Llama-3.1-8B (24.3s vs 30.5s). These reductions arise because only a minority of heads keep full sequences, while locality heads operate on much shorter caches. The variable-length FlashAttention API exploits this heterogeneity efficiently.

ContextKeeper substantially reduces KV memory and improves decoding throughput, while requiring only lightweight profiling to identify CA heads, providing consistent latency gains, demonstrating its practical utility under tight GPU memory budgets.

## 5 ABLATION STUDIES

### 5.1 EFFECT OF TASK SELECTION IN THE PROFILING SET

To identify the CA heads effectively, the profiling set should include a diverse range of tasks. We examine the impact of task selection by comparing summarization performance with and without the GovReport task in the profiling set (with overlapping samples in LongBench excluded). Removing GovReport leads to a slight performance decrease on both En.Sum (42.22 → 40.89) and GovReport (34.72 → 32.36). Nevertheless, the model still outperforms RazorAttention, which primarily captures surface-level induction or echo patterns. This suggests profiling must include at least one global-aggregation task (e.g., summarization) to reliably surface CA heads that attend beyond induction/echo patterns. It also highlights its flexibility, users may select tasks most relevant to their own deployment scenarios when constructing the profiling set.

## 6 CONCLUSION

In this work, we addressed the memory and latency bottlenecks of long-context LLM inference by proposing ContextKeeper, a training-free, head-specific KV retention policy. ContextKeeper identifies a minority of context-anchored (CA) heads that consistently attend to middle-context tokens across tasks and preserves their full KV cache, while dropping context tokens for locality heads. This design maintains multi-turn fidelity that query-centric methods lose, yet requires only lightweight inference-time profiling to identify CA heads. Experiments show that ContextKeeper achieves up to 3.86× KV memory reduction on MHA models and 1.25× decoding speedups on both MHA and GQA models, while retaining near-full accuracy on LongBench and about 89% of full performance on the much longer SCBench benchmark. Together, these results demonstrate that preserving context only where it matters — in a small subset of attention heads — offers a practical and plug-and-play approach to scaling LLM inference under tight memory budgets.

REPRODUCIBILITY STATEMENT

We will release code, the profiling set for CA identification, plus scripts to reproduce the results stated in this paper.

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

## A  APPENDIX

### A.1  THE USE OF LARGE LANGUAGE MODELS (LLMS)

We only use AI tools like ChatGPT to polish the written text.

### A.2  EXPERIMENT RESULTS ON MORE MODELS

During the rebuttal period, we further evaluated ContextKeeper on additional model families. The results demonstrate our method has strong generalizability across architectures and newer models. As shown in the table 5, on Qwen3-32B, ContextKeeper retains 99.25% of the full-KV performance at 56% sparsity. Similarly, on Qwen2.5-72B-Instruct, it preserves 99.01% of the full-KV performance while reaching 68% sparsity.

| Tasks | Qwen3-32B Full KV | Qwen3-32B Context(56%) | Qwen2.5-72B Full KV | Qwen2.5-72B Context(68%) |
|---|---|---|---|---|
| NrtvQA | 32.24 | 32.95 | 33.99 | 32.54 |
| Qasper | 47.71 | 46.61 | 51.01 | 48.28 |
| MF-en | 52.58 | 51.83 | 53.28 | 54.01 |
| HotpotQA | 51.84 | 52.00 | 65.84 | 65.92 |
| 2wikiMQA | 54.14 | 53.60 | 64.36 | 64.95 |
| MuSiQue | 27.63 | 26.17 | 46.00 | 44.88 |
| GovReport | 33.16 | 32.09 | 34.61 | 33.80 |
| QMSum | 24.04 | 24.14 | 23.77 | 23.46 |
| MultiNews | 25.33 | 25.01 | 24.93 | 24.61 |
| TREC | 76.00 | 74.00 | 74.50 | 75.00 |
| TriviaQA | 90.16 | 90.35 | 90.01 | 89.71 |
| SAMSum | 41.70 | 40.38 | 41.32 | 37.99 |
| PCount | 15.50 | 16.50 | 7.50 | 7.50 |
| PRe | 100.0 | 100.0 | 95.50 | 95.50 |
| LCC | 70.88 | 70.71 | 73.35 | 74.03 |
| RB-P | 67.20 | 67.63 | 74.02 | 73.25 |
| **Average** | 50.63 | 50.25 | 53.37 | 52.84 |

Table 5: Experiment results across tasks on LongBench with Llama-2-7B-32K-Instruct model.

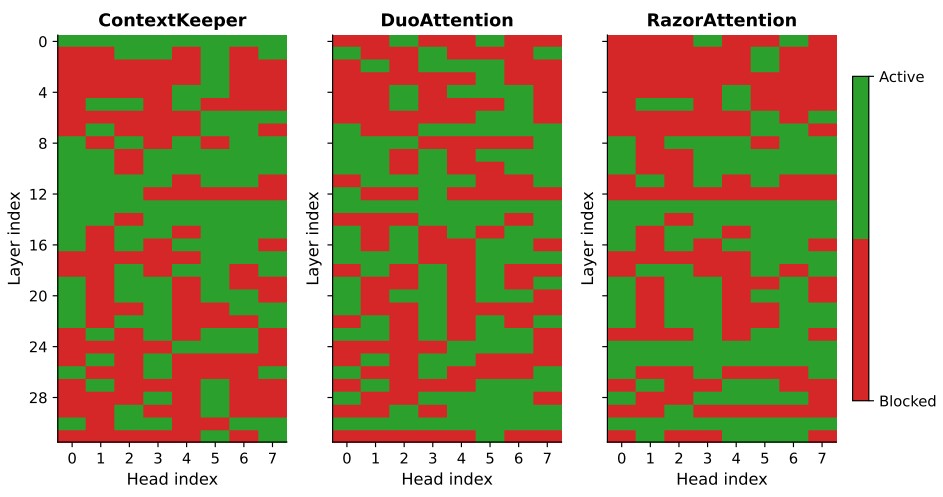

Figure 4: Comparison of blocked attention heads across ContextKeeper, DuoAttention, and RazorAttention. Heatmaps show which heads are blocked (red) or active (green) at each layer of Llama-3.1-8B-Instruct under a 50% sparsity setting. The patterns reveal that the three methods block notably different subsets of heads.

## A.3 BLOCKED ATTENTION HEADS ANALYSIS

To compare how different methods block attention heads, we evaluate ContextKeeper, DuoAttention, and RazorAttention on Llama-3.1-8B-Instruct, and apply a uniform sparsity rate of 50%.

Figure 4 shows that each method identifies a notably different subset of attention heads to block. To quantify these differences, Figure 5 reports the overlap of blocked heads across methods. We find that 73 out of 128 heads (57%) are blocked by all three methods. Beyond this shared core, ContextKeeper differs from RazorAttention by 20.3% and from DuoAttention by 28.9%.

These results demonstrate that ContextKeeper is not a simple renaming of DuoAttention's retrieval heads. Instead, ContextKeeper identifies heads based on stable mid-context attention patterns (echo and induction-like structures), which yields a selection distribution that is closer to RazorAttention

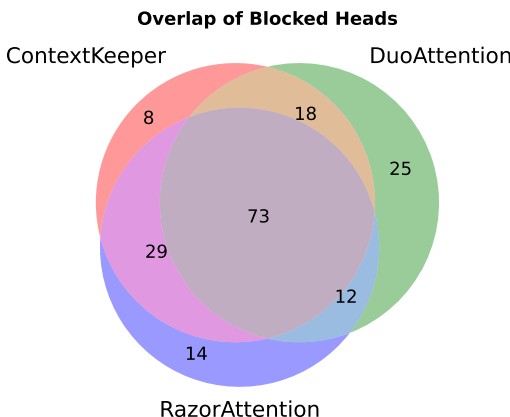

Figure 5: Venn diagram of blocked attention heads across ContextKeeper, DuoAttention, and RazorAttention. Among 128 heads, 73 are blocked by all three methods, while the remaining regions highlight each method's unique head selections, underscoring their differing blocking strategies.

but still distinct. ContextKeeper also detects more abstract contextual patterns not explicitly targeted by RazorAttention, resulting in a unique blocked-head profile.

