# OpenReview forum: "ContextKeeper: Head-Specific KV Cache Retention for Long-Context LLM Inference"
_ICLR.cc/2026/Conference — Submitted to ICLR 2026_

### Official Review · Reviewer_kU1K · 2025-10-23

**Soundness:** 3
**Presentation:** 3
**Contribution:** 3
**Rating:** 2
**Confidence:** 4

**Summary:**

This paper proposes ContextKeeper, a training-free, head-specific KV cache retention method for long-context LLM inference. The key observation is that attention heads specialize differently: only a small subset consistently attends to middle-context tokens, while most focus on sink or recent tokens. ContextKeeper identifies these Context-Anchored (CA) heads through a lightweight inference-only profiling procedure that measures each head’s attention to context tokens across a few tasks. During inference, it retains full KV caches only for CA heads and keeps reduced caches for others, implemented efficiently via a FlashAttention-compatible design. Experiments on Llama-2-7B and Llama-3.1-8B across LongBench and SCBench show up to 3.86× memory reduction and 1.25× decoding speedup with near-full accuracy, preserving multi-turn fidelity better than prior methods like RazorAttention and DuoAttention. The method offers a simple, practical, and scalable approach to efficient long-context LLM deployment.

**Strengths:**

1. ContextKeeper identifies CA heads entirely through inference-time profiling, requiring only a few samples and no retraining, which makes it plug-and-play for existing models.
2. Across multiple benchmarks (LongBench, SCBench) and two architectures (MHA, GQA), the method achieves substantial KV memory reduction (up to 3.86×) and decoding speedup (≈1.25×) with minimal accuracy degradation, outperforming prior methods such as RazorAttention and DuoAttention.

**Weaknesses:**

1. The overall concept of ContextKeeper—splitting attention heads into those with full KV caches and those with truncated ones, is not brand new. The proposed framework is still similar to existing work such as DuoAttention.
2. Despite large KV memory reduction, the reported decoding speedup (≈1.25×) is relatively small and may not translate into practical end-to-end efficiency gains in practical use cases, particularly the multi-batch or distributed inference settings.

**Questions:**

Could you provide a latency comparison between ContextKeeper and other KV compression methods such as DuoAttention, SnapKV, and RazorAttention under the same setup?

---

> ### Author Response · Authors · 2025-12-04
>
> Thanks for your detailed and constructive feedback! In the following, we provide a point-by-point clarification addressing each of your concerns.
>
> **W1: Novelty doubt about the ContextKeeper**
>
> We acknowledge that the high-level idea of applying different KV-cache retention strategies to different groups of attention heads has appeared in prior work. However, ContextKeeper introduces a distinct perspective on which heads should receive full retention and why.
>
> Existing methods such as DuoAttention define retrieval heads based on how much model output degrades under perturbations using training-based method. In contrast, ContextKeeper identifies Context-Anchored (CA) Heads by analyzing stable structural patterns in their attention distributions—specifically, their consistent anchoring to middle-context tokens that carry essential long-range information. This mid-context anchoring cannot be captured by the deviation-based training procedure used in DuoAttention and often differs from the induction- or echo-pattern criteria used in RazorAttention.
>
> As shown in our updated appendix, the head subsets selected by ContextKeeper and by DuoAttention diverge substantially, and the CA Heads we identify play a different functional role, particularly in summarization tasks requiring cross-segment reasoning or high level abstraction. This difference in principle, not just procedure, underlies the improved performance we observe on such tasks.
>
> Thus, while the overall framework of head-partitioned KV retention may appear similar at a macro level, the definition, motivation, and resulting behavior of the head groups are meaningfully different from prior work.
>
> **W2: Limited decoding speedup**
>
> Thank you for raising this point. We would like to clarify two aspects regarding the speedup results.
>
> - Source of speedup and practical impact.
>
> The reported 1.25× decoding speedup stems from the reduced KV-cache memory footprint after pruning non-CA heads. This directly lowers memory bandwidth requirements during decoding, reducing KV loading time. More importantly, the reduced KV memory consumption allows the model to handle more concurrent sequences or longer contexts within a fixed GPU memory budget—an efficiency gain that is not fully reflected in single-sequence speedup numbers but is highly relevant in production inference workloads.
>
> - Evaluation framework and compatibility with practical systems.
>
> Our speedup experiments were conducted using the transformers library, which does not incorporate many low-level optimizations used in high-throughput engines like vLLM. We are currently integrating ContextKeeper into vLLM, where the benefits are expected to translate more directly to real-world, latency-sensitive deployments. Additionally, the set of CA Heads is input-independent, meaning ContextKeeper is naturally compatible with multi-batch and distributed inference settings without per-request recomputation or dynamic head selection.
>
> We appreciate the reviewer’s suggestion and will include a discussion on these practical considerations and ongoing vLLM integration efforts in the final version.
>
> **Q1: Latency comparison**
>
> Thank you for the suggestion. We agree that latency is an important dimension of comparison. However, a direct, apples-to-apples latency evaluation across DuoAttention, SnapKV, RazorAttention, and ContextKeeper is not straightforward, for the following reasons:
>
> - SnapKV operates under a fundamentally different compression paradigm. SnapKV enforces a fixed token budget by dropping most context tokens—including tokens needed for future turns. In contrast, ContextKeeper preserves all tokens for CA Heads and prunes for other heads. Because the algorithms operate differently, a latency comparison under the same setup would not be methodologically fair or interpretable.
>
> - For DuoAttention and RazorAttention, a fair comparison requires using the same inference engine and identical KV-retention mechanics. Both methods also partition heads into two groups, but differ only in which heads are given full KV retention. Under a unified implementation, latency depends primarily on the effective sparsity ratio. As shown in our experiments, ContextKeeper consistently achieves higher sparsity at the same model-performance level, which directly implies lower KV-loading cost and thus lower decoding latency when implemented in the same framework.
>
> We thank the reviewer again for the thoughtful feedback and constructive suggestions, which will help further strengthen our manuscript.

---

### Official Review · Reviewer_1ay8 · 2025-10-28

**Soundness:** 3
**Presentation:** 3
**Contribution:** 2
**Rating:** 4
**Confidence:** 4

**Summary:**

This paper proposes ContextKeeper, a KV cache eviction strategy for LLM inference with long multi-turn contexts. The key claim is that different attention heads specialize: a small subset of “Context-Anchored (CA) heads” consistently attends to the middle of the conversation history across tasks, whereas most other heads primarily attend either the global sink tokens at the very beginning or the most recent tokens.

At inference time, ContextKeeper keeps the full KV cache only for those CA heads, while aggressively truncating KV for the other “locality heads,” which only retain sink + a rolling recent window.

The paper reports: up to ~3.86× KV memory reduction and ~1.25× decoding speedup, while maintaining ~89% of full-KV quality on SCBench, and near-parity on long-context benchmarks such as LongBench.

**Strengths:**

1. The paper emphasizes that identifying CA heads is “training-free”; it only requires running inference on a profiling set and computing attention statistics + consistency voting across samples, unlike methods that require optimizing gates or doing lightweight finetuning to classify retrieval heads.
2. The authors evaluate on SCBench (multi-turn / agent-style interactions) and ContextKeeper preserves ~90% of full-cache quality and often outperforms other KV-pruning baselines (including DuoAttention, Razor, etc.) in 7/9 categories, while still giving large KV savings.

**Weaknesses:**

1. Novelty vs. DuoAttention feels incremental:
- Conceptually, the method is extremely close to DuoAttention that (i) partitions attention heads into two groups, and (ii) keeps full KV history for the “important” group while aggressively pruning KV for the rest. DuoAttention refers to these as “retrieval heads” and “streaming heads,” and applies full-cache vs. sliding-window retention accordingly.
- This paper replaces “retrieval heads” with “Context-Anchored heads,” and replaces DuoAttention’s identification procedure with an offline profiling procedure that measures which heads consistently allocate attention scores to the middle of the context across a few evaluation tasks. The profiling is indeed more efficient to run, which is an advantage.
- However, at a high level, the policy is the same, while only the identification of heads with full KV cache changes. Thus, the contribution risks being viewed as an incremental variant of DuoAttention.
2. A main claim of the paper is that, unlike prior KV eviction algorithms, the proposed method preserves critical mid-context for later turns in multi-round dialogue. While the reported results are competitive against other compression baselines, the performance is still far below full attention on SCBench. In other words, the method is not yet "near-lossless" in the multi-turn regime it claims to specialize in.
3. DuoAttention explicitly reports gains not only in decode latency but also in prefill throughput, because it prunes the KV cache already during the prefilling. That matters a lot in long-context understanding scenarios where the prefill stage dominates end-to-end latency. By contrast, the proposed method mainly helps during decoding.

**Questions:**

1. How often do CA heads and DuoAttention’s retrieval heads actually disagree? Specifically, for a given model, what fraction of heads are selected by one method but not the other? Could you show a Venn diagram + ablation (“remove only the heads in the symmetric difference”) to prove that CA heads are not just a renaming?

---

> ### Author Response · Authors · 2025-11-29
>
> Thanks for your detailed and constructive feedback! In the following, we provide a point-by-point clarification addressing each of your concerns:
>
> **W1: Novelty doubt about the ContextKeeper.**
>
> Our method and DuoAttention both partition attention heads into two groups and apply different KV-cache retention strategies; however, the principles underlying head identification are fundamentally different, not incremental variations of each other.
>
> DuoAttention assumes that retrieval heads are those whose outputs degrade significantly when attention is restricted to recent tokens and attention sinks. Its identification procedure therefore relies on training-time perturbations and measuring output deviation.
>
> In contrast, ContextKeeper identifies Context-Anchored (CA) Heads based on structural attention-weight patterns that consistently point to the middle context, which carries the information required for long-context reasoning and follow-up queries. Our profiling procedure analyzes attention-weight distributions directly, revealing heads that systematically anchor to and retrieve from middle-context segments—even in cases where induction or echo patterns used by prior work (e.g., RazorAttention) fail. This leads to qualitatively different head selections, as shown in Figure 2, and enables superior performance on tasks such as summarization where sentence-boundary or semantic-anchor tokens are crucial.
>
> Moreover, CA-head identification is purely inference-based, requiring only minutes of profiling and no training, substantially improving scalability to very large models. DuoAttention, by contrast, incurs high GPU-memory costs, long training times, and non-trivial hyperparameter tuning.
>
> Thus, while both methods share the high-level idea of head partitioning, the core mechanism and the resulting behavior are different: ContextKeeper leverages structural, attention-driven token anchoring patterns that DuoAttention’s training-based retrieval-head definition does not capture.
>
> **W2: Not near-lossless performance on SCBench**
>
> Thank you for raising this point. In Section 4.3, we compare ContextKeeper with several strong baselines on SCBench. Consistent with the LongBench results, ContextKeeper outperforms all other compression baselines under the same sparsity setting. For fairness, we evaluated SCBench using the exact same sparsity ratio derived from LongBench profiling, and under this fixed setting, ContextKeeper preserves 88.9% of full-KV performance.
>
> We agree that this is not yet near-lossless. However, it is important to highlight that sparsity–performance is a tunable tradeoff. The chosen retention fraction in the current version reflects an aggressive sparsity level rather than the best possible fidelity. Users can achieve near-lossless accuracy by selecting slightly lower sparsity ratios, and our method supports this adjustment seamlessly.
>
> Due to time constraints during the rebuttal period, we have not yet completed SCBench experiments under varying retention fractions, which would more fully characterize the tradeoff curve and are expected to show near-lossless performance at moderate sparsity levels. We plan to include these results in the final version.
>
> **W3: Prefill throughput vs. decoding efficiency**
>
> We appreciate the reviewer raising this important distinction. Indeed, ContextKeeper is designed to optimize the decoding stage, and we do not claim improvements in prefill throughput. Our focus in this work is on accelerating long-context multi-turn interactions, where decoding overhead accumulates over many rounds.
>
> You raised a valuable point: extending CA heads based retention to the prefill stage may further reduce end-to-end latency for very long inputs. We consider this an important direction for future work and will test the compatibility with existing prefill-optimization techniques.
>
> **Q1: Disagreement between CA heads and retrieval heads**
>
> Thank you for the suggestion. We agree that quantifying the overlap between CA Heads and DuoAttention’s retrieval heads is important for demonstrating that the two are not equivalent. In the updated manuscript, we include a detailed comparison in the appendix: we visualize the head selections of both methods and provide further qualitative analysis. These results show that the two methods often select different subsets of heads, supporting our claim that CA Heads are not simply a renaming of retrieval heads.

---

### Official Review · Reviewer_mVMG · 2025-11-01

**Soundness:** 3
**Presentation:** 3
**Contribution:** 2
**Rating:** 4
**Confidence:** 4

**Summary:**

This paper presents ContextKeeper, a training-free, head-specific KV cache retention strategy for long-context LLM inference. The method builds on the observation that a small subset of attention heads, termed Context-Anchored (CA) heads, consistently attend to middle-context tokens, while most focus on sink and recent tokens. Preserving full caches only for CA heads maintains multi-turn fidelity and yields substantial memory and latency savings with minimal accuracy loss on standard long-context benchmarks.

**Strengths:**

1. Presents a lightweight, training-free identification procedure and a plug-and-play KV cache retention design that is compatible with FlashAttention.
2. Offers clear organization and writing, with motivation, methodology, and experimental results presented in a manner that is easy to follow.
3. Provides extensive experimental evidence showing that ContextKeeper consistently outperforms competitive baselines across diverse long-context benchmarks.

**Weaknesses:**

1. Lack of novelty, as similar head specialization concepts have been explored in RazorAttention and DuoAttention.
2. Insufficient discussion of the advantages of the proposed head identification method compared with DuoAttention.

**Questions:**

1. ContextKeeper and DuoAttention adopt similar strategies, both utilizing a synthetic dataset to identify different head types, and the definition of CA heads is closely aligned with that of retrieval heads. The primary distinction lies in the identification method. Could the authors clarify why ContextKeeper’s identification approach is preferable to DuoAttention’s training-based method? In my view, offline training does not impact inference efficiency. If dataset choice is the critical factor, could the authors train DuoAttention using the profiling dataset constructed for ContextKeeper and provide a direct comparison?
2. In identifying CA heads, you select the last $m$ tokens from the prefill stage and the first $k$ tokens from the decoding stage as the observation window. Could you elaborate on the rationale for this choice? Is there a risk that some tokens in this window attend heavily to context tokens while others do not, leading to skewed C-preference scores and potentially omitting relevant heads?
3. Have you evaluated ContextKeeper across different model architectures and varying retention fractions to provide a more comprehensive picture of its performance?

---

> ### Author Response · Authors · 2025-11-29
>
> Thanks for your detailed and constructive feedback! In the following, we provide a point-by-point clarification addressing each of your concerns.
>
> **W1/W2/Q1: Lack of novelty; Comparison between ContextKeeper and DuoAttention**
>
> We acknowledge that head-splitting strategies and retaining KV caches for a subset of heads have been explored in prior work. However, ContextHeads are not merely a renaming of retrieval heads. Although they appear related, the underlying identification mechanisms are distinct.
>
> DuoAttention identifies retrieval heads by measuring output deviations through additional training, while RazorAttention detects important heads using induction and echo patterns observed in attention weights. Our method is closer in spirit to RazorAttention in that it leverages attention-weight analysis, but our design enables it to locate more different structural patterns. As shown in Figure 2, tokens such as sentence-boundary markers (e.g., [END]) cannot be reliably captured through induction or echo patterns, whereas our method successfully identifies them. This distinction enables ContextKeeper to outperform RazorAttention on summarization tasks.
>
> To answer your questions directly:
>
> > Could the authors clarify why ContextKeeper’s identification approach is preferable to DuoAttention’s training-based method? In my view, offline training does not impact inference efficiency.
>
> ContextKeeper identifies long-context–critical heads using only a few minutes of inference, with negligible overhead beyond standard model execution. This makes the method highly practical and scalable.
>
> In contrast, DuoAttention requires substantial GPU memory, long training durations, and tuning multiple hyperparameters, especially for large models such as Qwen2.5-72B-Instruct. While it is true that offline training does not affect inference latency, ContextKeeper achieves comparable or better sparsity–performance trade-offs with far fewer resources and in a significantly shorter time.
>
> > If dataset choice is the critical factor, could the authors train DuoAttention using the profiling dataset constructed for ContextKeeper and provide a direct comparison?
>
> Dataset choice is NOT the critical factor. Training DuoAttention on our profiling dataset might yield slightly stronger results, but it would still require thousands of training steps, and both the training time and the number of trainable parameters scale with model size. In contrast, ContextKeeper consistently requires only inference-level resources and finishes profiling within minutes, regardless of model scale.
>
> **Q2: Explanation of token selection during CA heads identification.**
>
> We select the last $m$ tokens from the prefill stage because these tokens typically correspond to the query portion of the user prompt. This design is analogous to SnapKV’s observation window, which focuses on the final segment of the prompt. As demonstrated in SnapKV, even when the last m tokens do not explicitly contain the query, the attention patterns within this window still provide reliable signals for identifying important tokens.
>
> For the first $k$ tokens from the decoding stage, our empirical observation is that during generation, the model consistently attends to tokens that are critical for answering the current query. Therefore, collecting the first $k$ decoding steps allows us to analyze which heads are essential for long-context reasoning.
>
> After profiling, we aggregate attention weights across all tokens in this window and select heads with the highest cumulative weights. We further apply voting across all samples within a task and then across tasks. This multi-level aggregation mitigates bias and prevents skewed C-preference scores, ensuring that crucial heads are not overlooked.
>
> **Q3: Additional evaluations on different architectures and varying retention fractions**
>
> During the rebuttal period, we conducted additional experiments on more model families. The results indicate that ContextKeeper generalizes well across architectures and newer model generations. As reported in the updated appendix, on Qwen3-32B, ContextKeeper retains 99.25% of the full-KV performance at 56% sparsity. Similarly, on Qwen2.5-72B-Instruct, it preserves 99.01% of the full-KV performance while reaching 68% sparsity.
>
> Due to time constraints, we have not yet completed experiments on varying retention fractions. We plan to include these results in the final version.
>
> We thank the reviewer again for the thoughtful feedback and constructive suggestions, which will help further strengthen our manuscript.

---

### Official Review · Reviewer_2Y2t · 2025-11-01

**Soundness:** 3
**Presentation:** 2
**Contribution:** 2
**Rating:** 4
**Confidence:** 4

**Summary:**

This paper proposes ContextKeeper, a training-free KV cache compression method. ContextKeeper identifies a subset of attention heads—Context-Anchored (CA) heads—that consistently attend to middle-context tokens. By retaining full KV cache for CA heads while dropping middle tokens for others,  ContextKeeper can reduce memory and latency while preserving performance.

**Strengths:**

- Minimal Accuracy Drop: Experiments on LongBench show that ContextKeeper reduces KV cache size by up to 3.86× with negligible average accuracy loss, showing effective compression without sacrificing general capability.

- ContextKeeper is training free with minimal cost. The CA head identification requires only ~10 minutes of inference, making it highly practical.

**Weaknesses:**

- The experiments are unconvincing: Experiments are conducted only on Llama-2-7B-32K and Llama-3.1-8B-Instruct. Llama-2-7B-32K is outdated with a small context window and poor long-context ability compared to SOTA models. Testing on more architectures and newer models would strengthen generalizability. (e.g. Llama-3.1-8B-1024K, Phi-3-Mini-128K, Qwen2.5/3, etc).
- No experiments on extremely long context settings: Currently, ContextKeeper lacks scalability to extremely long context settings. The RULER benchmark should be tested, since it is a more accepted dataset for long-context scenarios with context lengths up to 256k.
- Keeping the KV cache for certain heads is not a novel idea. Similar head classification methods have been proposed in recent works like  DuoAttention. The author should discuss the novelty of ContextKeeper compared with these papers.

**Questions:**

The appendix is empty. Why not include more experimental details in the appendix?

---

> ### Author Response · Authors · 2025-11-29
>
> Thanks for your detailed and constructive feedback! Please find our point-by-point response below.
>
> **W1: The experiments are unconvincing.**
>
> During the rebuttal period, we further evaluated ContextKeeper on additional model families. The results demonstrate strong generalizability across architectures and newer models. As shown in the following table, on Qwen3-32B, ContextKeeper retains **99.25%** of the full-KV performance at **56% sparsity**. Similarly, on Qwen2.5-72B-Instruct, it preserves **99.01%** of the full-KV performance while reaching **68% sparsity**.
>
> |  | Qwen3-32B Full KV | Qwen3-32B Ours (56% Sparsity) | Qwen2.5-72B Full KV | Qwen2.5-72B Ours (68% Sparsity) |
> |:-----------|:------------:|:------------:|:------------:|:------------:|
> NrtvQA            | 32.24 | 32.95 | 33.99 | 32.54 |
> Qasper            | 47.71 | 46.61 | 51.01 | 48.28 |
> MF-en             | 52.58 | 51.83 | 53.28 | 54.01 |
> HotpotQA          | 51.84 | 52.00 | 65.84 | 65.92 |
> 2wikiMQA          | 54.14 | 53.60 | 64.36 | 64.95 |
> MuSiQue           | 27.63 | 26.17 | 46.00 | 44.88 |
> GovReport         | 33.16 | 32.09 | 34.61 | 33.80 |
> QMSum             | 24.04 | 24.14 | 23.77 | 23.46 |
> MultiNews         | 25.33 | 25.01 | 24.93 | 24.61 |
> TREC              | 76.00 | 74.00 | 74.50 | 75.00 |
> TriviaQA          | 90.16 | 90.35 | 90.01 | 89.71 |
> SAMSum            | 41.70 | 40.38 | 41.32 | 37.99 |
> PCount            | 15.50 | 16.50 | 7.50  | 7.50  |
> PRe               | 100.0 | 100.0 | 95.50 | 95.50 |
> LCC               | 70.88 | 70.71 | 73.35 | 74.03 |
> RB-P              | 67.20 | 67.63 | 74.02 | 73.25 |
> **Average**       | 50.63 | 50.25 | 53.37 | 52.84 |
>
> **W2: No experiments on extremely long context settings.**
>
> We use SCBench to evaluate performance under extremely long-context settings. Across the nine tasks we select from SCBench, the average input length is 117K tokens, which is close to the 128K maximum context length of the Llama-3.1-8B-Instruct model. This makes SCBench a suitable benchmark for assessing behavior in near-capacity long-context scenarios.
>
> SCBench contains several tasks that are similar or identical to those in RULER—such as key-value retrieval and multi-hop tracing—but it additionally supports multi-turn evaluations. This aligns well with our objective: ContextKeeper preserves all intermediate context tokens and maintains multi-turn fidelity, enabling accurate follow-up responses that depend on earlier segments of extremely long inputs.
>
> **W3: Keeping the KV cache for certain heads is not a novel idea.**
>
> We acknowledge that prior work has explored retaining KV caches for a subset of attention heads. Our contribution lies in identifying Context-Anchored (CA) Heads during inference, whereas existing training-based approaches—such as DuoAttention—identify retrieval heads through additional training procedures. For large models like Qwen2.5-72B-Instruct, the training-based strategy in DuoAttention requires substantially more GPU memory and significantly longer training time. In contrast, our method identifies CA Heads using only a few minutes of inference, making the process far more practical and resource-efficient.
>
> **Q1: The appendix is empty.**
>
> Due to the tight timeline before the initial submission, we were unable to include additional experimental results and analyses in the appendix. We have now added these materials for completeness.
>
> We thank the reviewer again for the thoughtful feedback and constructive suggestions.

---

### Meta-Review · Area_Chair_s5Wu · 2026-01-05

**Summary:**

The submission studied the KV management technique in long-context LLM inference, which is an important problem to save the cost. Different from previous KV cache methods that reduce tokens irrelevant to the current query, the authors focus on the head-specific personalization to deal with token KV. Experimental results demonstrate the performance of the proposed method ContextKeeper.

**Reviewer Concerns:**

Generally, the reviewers' concerns mainly focus on the following points:

1) The novelty is limited, as the head-specific KV design have been explored in RazorAttention and DuoAttention although they may introduce training.
2) The experiments are unconvincing due to the experimental setup for some outdated models with poor long-context ability, and the limited performance gain. Besides, ruler bench that is widely adopted in previous works has not tested.
3) Some writing issues make the submission not self-contained or well demonstrated, like missing experimental details, insufficient discussions etc.

**Reviewer Scores:**

Four reviewers respectively rated the submission 4, 4, 4, 2. According to the authors' rebuttal, I think parts of the concerns can be addressed. For example, the authors provided more experiments with recent open-source models with long-context ability like Qwen-3 series model. But regarding the novel point, AE does not think the reviewers will change their minds given the limited differences in head selection, and also lack of the experiments in ruler bench and lack of comparison with the Duoattention.

---

### Decision · Program_Chairs · 2026-01-26

Reject